# Prevalence of Cancer in Acid Sphingomyelinase Deficiency

**DOI:** 10.3390/jcm10215029

**Published:** 2021-10-28

**Authors:** Wladimir Mauhin, Thierry Levade, Marie T. Vanier, Roseline Froissart, Olivier Lidove

**Affiliations:** 1Internal Medicine Department, Groupe Hospitalier Diaconesses Croix Saint Simon, 75020 Paris, France; olidove@hopital-dcss.org; 2Metabolic Biochemistry Laboratory, Reference Center for Hereditary Metabolic Disorders, Biology Institute, Toulouse University Hospital, 31000 Toulouse, France; thierry.levade@inserm.fr; 3Toulouse Cancer Research Center, INSERM-University of Toulouse Paul Sabatier, 31000 Toulouse, France; 4INSERM and Hôpitaux de Lyon, 69002 Lyon, France; vaniermtv@gmail.com; 5Biochemical and Molecular Biology Department, Lyon University Hospital, 69500 Bron, France; roseline.froissart@chu-lyon.fr

**Keywords:** acid sphingomyelinase deficiency, cancer, ceramide, lysosomal storage disease, Niemann–Pick disease

## Abstract

Acid sphingomyelinase deficiency (ASMD) is an inherited lysosomal disease characterised by a diffuse accumulation of sphingomyelin that cannot be catabolised into ceramide and phosphocholine. We studied the incidence of cancer in ASMD patients. We retrospectively reviewed the medical records of the adult chronic visceral ASMD patients in our cohort. Thirty-one patients (12 females, 19 males) were included with a median age of 48.7 y. (IQ: 30.3–55.1). Five cancers were observed in 1 female (breast cancer) and 4 males (two lung cancers, one thyroid cancer and one bladder cancer), resulting in a prevalence of 16.1%. The existence of cancer was associated with a more severe ASMD characterised by a larger spleen (25 cm (22.5–25) vs. 18 cm (17–20); *p* = 0.042); lower diffusing capacity of the lung for carbon monoxide (DLCO; 29.5 % (17.8–43.0) vs. 58.5 % (49.8–69.5%); *p* = 0.01) and tobacco use (100% vs. 45%; *p* = 0.04). Three patients died, all from cancer (*p* = 0.002). The prevalence of cancer appeared to be strikingly elevated in our cohort of patients, without any specificity in the type of cancer. Systematic screening for cancer should be performed, and carcinogenic substances such as tobacco should be avoided in patients with ASMD.

## 1. Introduction

Acid sphingomyelinase deficiency (ASMD; OMIM# 607616), also known as Niemann–Pick disease types A and B (NPD-A and B), is a rare autosomal recessive lysosomal storage disease with an estimated birth prevalence of 0.4–0.6/100,000 [1], a figure that is likely underestimated. Acid sphingomyelinase (ASM) is a lysosomal lipid hydrolase encoded by the *SMPD1* gene (locus 11p15.4) [2]. Due to an enzymatic defect in ASM, a progressive and diffuse accumulation of sphingomyelin occurs in all cells and tissues, primarily in reticuloendothelial and hepatocyte cells, but also in the neurons of neuronopathic types [2,3]. The degree of in vivo residual enzymatic activity leads to a clinical continuum in terms of severity [2,4,5,6]. The absence of residual ASM activity leads to the most severe phenotype, the infantile neurovisceral ASMD (NPD-A), characterised by a rapidly progressive condition associating neurodegeneration, hepatomegaly, failure-to-thrive and death by 4 years of age [2]. In contrast, a higher residual ASM activity is associated with chronic visceral ASMD (CV-ASMD or NPD-B), and its diagnosis can vary from infancy to adulthood. CV-ASMD is characterised by hepatosplenomegaly, secondary anaemia, thrombocytopenia and interstitial lung disease (ILD) [2,4,5]. In recent years, an unexpected diagnosis of cancer was observed in several CV-ASMD patients referred to our tertiary centre. Malignancies have been described sporadically in ASMD patients without any specific warning [4,5]. However, the role of ASM deficiency in cancer development has already been suggested [7]. Therefore, we decided to evaluate the prevalence of cancer in our cohort of adult CV-ASMD patients.

## 2. Patients and Methods

We performed a retrospective review of the medical records of all the CV-ASMD patients who were referred to our tertiary centre in Paris, France, from January 2011 to July 2021. Patients older than 18 years of age with enzymatically or genetically proven diagnosis of ASMD were included.

The clinical assessment made during the last follow-up visit or at the diagnosis of cancer was considered. Spleen diameter was assessed by abdominal ultrasound echography. Splenomegaly was defined as a spleen diameter of >14 cm. ILD severity was assessed with the carbon monoxide diffusing capacity of the lung (DLCO) adjusted for haemoglobin, performed by standard clinical techniques. Abnormal DLCO was defined as <80%. Adrenal gland abnormality was defined as a hypertrophic gland ± nodule observed on an abdominal CT scan. Consanguinity status, tobacco and alcohol exposure (past or ongoing) were defined according to the patient reports. Missing data were not included in the statistics.

Statistical analyses were performed with GraphPad Prism 8 software^®^ (San Diego, CA, USA) and the EZR package for R [8]. Fisher’s exact *t*-test and logistic regression were used to assess the associations between specific conditions and cancer. The Mann–Whitney test was used to compare quantitative data between cancer and non-cancer groups. Kaplan–Meier survival analysis was performed where survival probability is calculated as the number of patients surviving divided by the number of patients at risk. Acid sphingomyelinase activities in leukocytes were assayed using a radiolabelled sphingomyelin substrate [9]. This study was approved by a local ethics committee.

## 3. Results

### 3.1. Patients

Thirty-one patients (12 females, 19 males) from 28 pedigrees were included in this study with a median age at the last visit of 48.7 years (interquartile range (IQ): 30.3–55.1). The median age at diagnosis was 8.1 years (IQ: 1.8–42.3). The median body mass index was 24.2 kg/m^2^ (IQ: 21.5–27.6). Consanguinity was reported by 16 of the 31 patients. All the patients exhibited splenomegaly with a median diameter of 18 cm (IQ: 17–22). Two patients had a splenectomy in the past. Among the 24 patients with available DLCO values, 87.5% had DLCO <80%. Median DLCO was 51.5% (IQ: 42–69). Five cancers were observed in 1 female and 4 males, all after the diagnosis of ASMD. The most prevalent *SMPD1* genotype was p.R610del in homozygosity observed in 14 of the 29 patients with available information. The mean residual ASM activity in patients with peripheral leucocytes was around 15% of the normal. Ten patients had abnormal serum protein electrophoresis (among 23 available), with 5 having polyclonal hypergammaglobulinemia (all between 15 and 20 g/L), 2 having hypogammaglobulinemia (6–8 g/L), 2 having monoclonal gammopathies of undetermined significance (IgA and IgG) and one having type 2 cryoglobulinemia. The characteristics of the patients are listed in Table 1.

### 3.2. Description of Cancers

#### 3.2.1. Lung Cancers (*n* = 2)

A metastatic neuroendocrine small-cell lung carcinoma was diagnosed in a 58.2 year old former smoker male patient (30 pack-years) with the *SMPD1* genotype p.R610del/p.R610del. It is noteworthy that the patient was splenectomised because of a splenic rupture at the age of 52. Despite an initial good response to radiation therapy and chemotherapy, the evolution rapidly worsened. The patient died after 8 months of evolution.

A typical and metastatic well-differentiated squamous cell carcinoma was diagnosed in a 57.7 year old smoker male patient (5 pack-years) with the *SMPD1* genotype p.R610del/p.R610del. Unfortunately, the patient died 6 months after the diagnosis.

#### 3.2.2. Bladder Cancer (*n* = 1)

A non-metastatic high-grade urothelial carcinoma of the bladder was diagnosed in a 52.3 year old former smoker male patient (50 pack-years) with the *SMPD1* genotype p.R610del/p.R610del. The oncological therapeutic options were limited due to chronic respiratory failure and large hepatosplenomegaly with portal hypertension and severe thrombocytopenia due to the ASMD and chronic exposure to alcohol. Unfortunately, the patient died after 2 years of evolution.

#### 3.2.3. Papillary Thyroid Carcinoma (*n* = 1)

A papillary thyroid carcinoma (pT1bm N0 R0 V0 according to current classification) was diagnosed in a 40.3 year old former smoker male patient (15 pack-years) with the *SMPD1* genotype p.G247S/p.R476W. Thyroidectomy and radioactive iodine therapy (I-131) were performed with success. The cancer is considered in remission, and the patient is alive after 2 years of follow-up.

#### 3.2.4. Breast Cancer (*n* = 1)

A non-metastatic infiltrating ductal carcinoma was diagnosed in a 41.2 year old former smoker female patient (60 pack-years) with the *SMPD1* genotype p.Q134X/p.G244R. It is noteworthy that the patient, diagnosed at 24 years of age, was splenectomised 5 years later because of thrombocytopenia. The cancer is now considered cured, and the patient is alive after 10 years of follow-up.

### 3.3. Conditions Associated with Cancer

The characteristics of patients with or without cancer are presented in Table 2. Using non-parametric Fisher exact *t*-tests, we identified that splenectomy and exposure to tobacco were associated with the occurrence of cancer (respectively 40 vs. 0%; *p* = 0.02 and 100% vs. 43.5%; *p* = 0.04). Exposures to alcohol, gender and consanguinity status were not associated with the occurrence of cancer. Three patients from our cohort died from 2011, all from cancer. Therefore, death was statistically associated with cancer (*p* = 0.002).

Using non-parametric Mann–Whitney tests, we observed that the spleen size was larger in patients with cancer compared to patients without cancer (25 vs. 18 cm; *p* = 0.04, *n* = 22 data available) and that DLCO was significantly lower (29.5 vs. 58.5%; *p* = 0.01, *n* = 24 data available). There was a trend for higher body mass index (BMI) in patients with cancer (25.7 vs. 24.0 kg/m^2^; *p* = 0.085). There was no difference in terms of C-reactive protein, HDL or LDL cholesterol, haemoglobin or platelet levels. Plasma protein electrophoresis showed three monoclonal disorders (two MGUS (monoclonal gammopathy of undeteminerd significance) and one cryoglobulinemia), representing 13.0% of the cohort; spleen size and DLCO were not different in these patients compared to others. ASM residual activities were not different between patients with cancer compared to patients without (*p* = 0.4).

Using univariate regression analyses, DLCO values were inversely correlated to cancer occurrence with an odds ratio (OR) of 9 × 10^−7^ (*p* = 0.04). The correlation between spleen size and the occurrence of cancer tended toward significance, with an OR of 1.62 (*p* < 0.06). Age, body mass index, C-reactive protein, cholesterol and ASM residual enzymatic activity were not correlated with cancer occurrence. In a multivariate regression analysis, DLCO and spleen size did not correlate with cancer occurrence, suggesting that DLCO and spleen size are correlated to each other.

### 3.4. Survival Analyses

In a Kaplan–Meier model (Figure 1), the median survival age without cancer (Figure 1A) was 58.2 years in our cohort. The median survival age (Figure 1B) was not reached.

## 4. Discussion

The prevalence of cancer (16.1%; 5 in 31 patients) appeared abnormally elevated in our cohort of CV-ASMD patients, compared to the general population. Indeed, the Global Cancer Observatory platform developed by the International Agency for Research on Cancer estimates that in 2020 the prevalence of cancer in France was about 2.3%. Although it was not identified as a major issue, cancer had already been reported in ASMD [4,5,7]. Sabourdy et al. reported a marginal zone lymphoma in an ASMD male aged 65 years and discussed the possible pro-oncogenic role of the ASM deficiency [7]. In an analysis of the causes of death of 52 CV-ASMD patients, 5 (9.6 %) presented a cancer between 43 and 65 years of age (2 liver cancers, 1 multiple myeloma, 1 chondrosarcoma, 1 unknown) [4]. In a natural history study from McGovern et al., one patient presented with liver cancer [5]. The elevated incidence of cancer in our cohort compared with existing reports may be explained by the older age of our patients, with a median age of 48.7 years (30.3–55.1) at the last follow-up. In both previous studies, patients were younger: the median age of CV-ASMD patients was 23.5 y. (range 0.58–72) in the study from Cassiman et al., while the cohort studied by McGovern et al. included only nine patients aged over 40 [4,5]. In our cohort, the occurrence of cancer was associated with larger spleen size and lower DLCO values. Spleen enlargement and low DLCO are markers of severity in CV-ASMD [2,10,11]. We further observed that cancer was significantly associated with splenectomy, already suggested to be associated with disease worsening [5,12]. It therefore appears that cancer would preferentially occur in severely affected patients. Three of the five cancers observed in our cohort were associated with the R610del variant that has been associated with a milder phenotype of ASMD, questioning the correlation between ASMD severity and cancer development [5]. Regarding the severity of the disease in these three patients, we believe that our hypothesis linking ASMD severity and cancer risk remains valid.

Conventional in vitro measurements of acid sphingomyelinase activity, even assayed with natural substrate, do not accurately reflect in vivo activity [13]. Conversely, there is a recent indication that the continuous spectrum of clinical manifestations in ASMD could be related to plasma lyso-sphingomyelin levels [13]. Hence, cancer might occur in patients with higher concentrations of this biomarker, for which only sporadic data were available in our cohort, not allowing for a comparative study.

It has been demonstrated that acid sphingomyelinase and ceramide levels play major roles in carcinogenesis [14,15]. Sphingomyelin is a major structural component of all plasma membranes. Physiologically, ASM catabolises the hydrolysis of sphingomyelin into ceramide and phosphocholine [2]. Ceramide can, in turn, be hydrolysed into sphingosine by ceramidase [16]. Ceramide and sphingosine can be phosphorylated respectively into ceramide-1-phosphate (C1P) by ceramide kinase, and sphingosine-1-phosphate (S1P) by sphingosine kinase 1 or 2 (SPHK 1 and 2) [16]. All of these reactions are summarised in Figure 2. Ceramide, sphingosine, C1P and S1P are all bioactive lipids [14,15,16]. Ceramide and sphingosine have pro-apoptotic effects, whereas C1P and S1P have pro-proliferative effects [14,15]. Cancer development depends on the carcinogenesis itself, but also on the antitumor immunity and the tumour microenvironment. The ceramide profile seems to influence all three of these mechanisms [14]. It has been demonstrated that the knockdown of SHPK1 leads to a decrease in S1P levels, an increase in ceramide and an inhibition of cancer cell survival [15]. Osawa et al. described that tumour growth was increased in ASM-deficient mice with reduced macrophages within the tumour [17]. Additionally, in ASM-deficient mice, it was demonstrated that neutrophil apoptosis was delayed compared to wild-type mice [18]. ASM was also shown to play a major role in the regulation of immune CD4 and CD8+ T cells, macrophages and natural killer cells to abrogate cancer cell viability [19,20]. ASM also allows an antitumor response through Th17 cells and diminishes the Treg response [21,22]. Therefore, one could hypothesise that a decrease in ceramide level due to acid sphingomyelinase deficiency with conserved SHPK and ceramide kinase activities would lead to an imbalance in favour of S1P and C1P and a subsequent pro-tumour effect. Whether lyso-sphingomyelins play a role in cancer development remains to be determined. Monitoring C1P and S1P levels and ceramidase activity may help to understand these mechanisms in CV-ASMD.

In our cohort, we observed three cases with monoclonal dysglobulinemia. As already mentioned, Cassiman et al. reported one case of multiple myeloma [4]. A higher incidence of dysglobulinemia and multiple myeloma has been described in Gaucher disease, another inherited lysosomal storage disease characterised by an excess in glucosylceramide and lyso-glucosylceramide (lyso-GL1) [23]. In Gaucher disease, gammopathies have been related to the development of clonal anti-lyso-glucosylceramide (LGL1) antibodies due to the activation of LGL1-specific CD1d-restricted invariant natural killer T cells (iNKTs) [24]. On the contrary, a decrease in iNKT cell levels and defective antigen presentation have been reported in ASM-deficient mice and ASMD patients [25]. Whether lyso-sphingomyelins play a role in the development of monoclonal gammopathies remains unknown. However, monitoring plasma protein electrophoresis may be of interest in ASMD patients.

No specific curative treatment is yet available for ASMD. However, an enzyme-replacement therapy using olipudase alfa showed promising results in different clinical trials. A reduction of liver and spleen volumes (−35.6 and −47.3% from baseline, respectively) and an improvement of DLCO (+35% from baseline) were observed after 30 months of treatment [26]. Studies after 42 months further showed a clearance of hepatic sphingomyelin and a normalisation of the lipid profiles [27]. The normalisation of the lipid profiles may hopefully limit the possible higher risk of cancer.

We observed that tobacco use was significantly associated with the occurrence of cancer. Smoking habits were frequent in our cohort. To better understand the role of ASM deficiency in carcinogenesis, it would be necessary to confirm these findings in a larger cohort and compare them to a smoker population. However, because of the possible higher susceptibility to cancer in ASMD patients, quitting tobacco and limiting all carcinogenic exposure appears to be a priority in the management of ASMD patients. Although this study was limited by the small number of patients, the high prevalence of cancer is of major concern. However, there should be special attention given to cancer screening in ASMD patients.

## 5. Conclusions

We observed an abnormally elevated incidence of cancers in our CV-ASMD adult patients. The severity of the disease appeared correlated with the risk of cancer. A biochemical rationale has already been reported, involving the imbalance of ceramide and its metabolites and the role of ASM in cancer development. It now appears essential to carefully screen cancer in CV-ASMD.

## Figures and Tables

**Figure 1 jcm-10-05029-f001:**
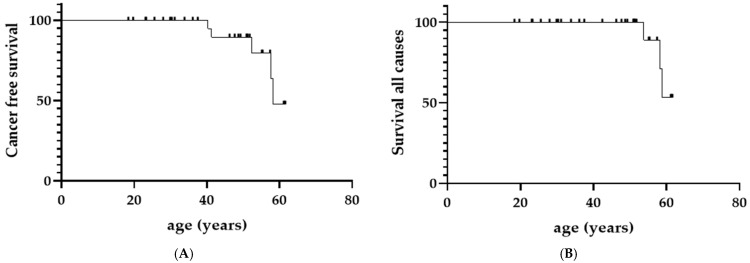
Kaplan–Meier survival probability curve in CV-ASMD (chronic visceral ASMD). (**A**) Survival age without cancer (*n* = 5 events); (**B**) Survival age considering all causes of death (*n* = 3 events).

**Figure 2 jcm-10-05029-f002:**
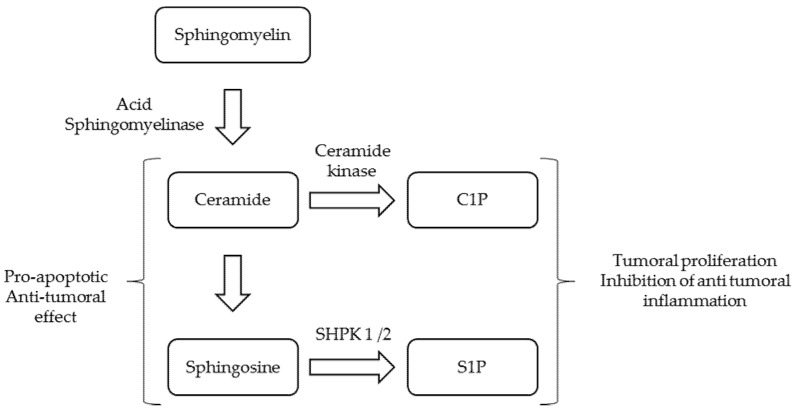
The role of ceramide and its derivatives in cancer development (C1P: ceramide-1-phosphate; S1P: sphingosine-1-phosphate; SPHK: sphingosine kinase).

**Table 1 jcm-10-05029-t001:** Characteristics of CV-ASMD patients (ASM: acid sphingomyelinase).

		Data Available, *n*
**Total, *n***	31	
**Males, *n***	19	
**Females, *n***	12	
**Peripheral leucocyte ASM activity (Median, IQ)**	0.36 nmol/h/mg protein (IQ: 0.21–0.55)	24
**Age at diagnosis (Median, IQ)**	8.1 y. (1.8–42.3)	31
**Age at last follow-up (Median, IQ)**	48.7 y. (30.3–55.1)	31
**Body mass index (Median, IQ)**	24.2 kg/m^2^ (21.5–27.6)	31
**Consanguinity**	51.6%	31
**Adrenal gland abnormality**	72.2%	18
**Spleen size (Median, IQ)**	18 cm (17–22)	23
**DLCO (Median, IQ)**	51.5% (IQ: 42–69)	24
**Haemoglobin (Median, IQ)**	14.3 g/dL (13.7–14.9)	26
**Platelets (Median, IQ)**	130 g/L (98–140)	31
**HDL cholesterol (Median, IQ)**	0.25 g/L (0.18–0.30)	21
**Tobacco exposure**	53.6%	28
**Alcohol exposure**	42.3%	26
**Cancer**	5	31
**Death**	3	31

CV-ASMD: chronic visceral ASMD; DLCO: carbon monoxide diffusing capacity of the lung; IQ: interquartile range; HDL: high-density lipoprotein.

**Table 2 jcm-10-05029-t002:** Characteristics of patients with or without cancer (BMI: body mass index).

	Cancer	No Cancer	*p*-Value
** *n* **	5	26	
**Gender male/female**	4/1	15/11	*0.6*
**Consanguinity (*n*, %)**	3 (60%)	13 (50%)	*1*
**Median age at last visit (IQ) (years)**	53.8 (51.4–57.7)	46.9 (30.0–51.7)	*0.2*
**BMI (kg/m^2^)**	25.7 (25.2–29.4)	24.0 (21.4–26.9)	*0.09*
**Spleen diameter (cm)**	25 (22.5–25)	18 (17–20)	*0.04*
** *n available data* **	*3*	*19*	
**Platelets (g/L)**	197 (91–274)	131 (103–146)	*0.5*
**DLCO (%)**	29.5 (17.8–43.0)	58.5 (49.8–69.5)	*0.01*
** *n available data* **	*4*	*20*	
**HDL cholesterol (g/L)**	0.27 (0.18–0.29)	0.27 (0.20–0.32)	*0.7*
** *n available data* **	*5*	*16*	
**Tobacco use *n* (%)**	5/5 (100)	10/23 (43.5)	*0.004*
**Alcohol use *n* (%)**	2/5 (40)	9/20 (45)	*1*
**Death *n* (%)**	3 (60)	0	*0.002*

## Data Availability

The data presented in this study are available on request from the corresponding author. The data are not publicly available due to the need to inform patient previously to share data.

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
