# Peer review of "Prevalence of Cancer in Acid Sphingomyelinase Deficiency"

_jcm, 2021, doi:10.3390/jcm10215029_

Round 1
Reviewer 1 Report
The manuscript entitled Prevalence of cancer in Acid Sphingomyelinase Deficiency by Wladimir Mauhin et al. reports a retrospective study of adult patients suffering from chronic visceral acid sphingomyelinase deficiency (CV-ASMD). Various clinical parameters were recorded and correlated with the occurrence of cancer (a total of 5 cancers within 31 patients).
The authors report for the first time that CV-ASMD patients have a higher than average risk of developing cancer.
The manuscript is written in a clear and scientific style and the study is well performed.
My only criticism is the lack of a comparison of the CV-ASMD patient cohort suffering from cancer to a tumor cohort of patients without Niemann-Pick disease type B or, even better, a cohort of tumor patients who are or were all smokers. It is prominent that all five tumor patients in the current study are either smokers or were smokers. Therefore, a comparison to the entire French population, in which the prevalence of developing cancer is 2.3%, is not entirely correct. However, the present data are supported by numerous laboratory studies in mice.
In my opinion, an important publication from 2013 is missing, in which Y. Osawa and colleagues described increased tumor growth in mice suffering from Asm deficiency [Osawa Y, Suetsugu A, Matsushima-Nishiwaki R, Yasuda I, Saibara T, Moriwaki H, Seishima M, Kozawa O. 2013. Liver acid sphingomyelinase inhibits growth of metastatic colon cancer. J Clin Invest 123:834-43].
Author Response
We thank the reviewer 1 for this review.
We agree with the reviewer 1 that in terms of statistics the CV-ASMD cohort should have been compared with a cohort of non-ASMD people with similar general characteristics and especially similar smoking habits or alcohol exposure. Our retrospective cohort-study is a preliminary work that could not be tailored to achieve this scale. However, the prevalence of cancer (16 %) appears particularly high even in a population of smoker. We have added this point in the limitations of our study.
We include the paper from Osawa et al. in our discussion.
Reviewer 2 Report
The Authors provided a single-center retrospective study on the prevalence of cancer in ASMD patients. This is a very interesting study that expands our knowledge regarding the natural history of the disease.
However, the Results of the study are quite intriguing.
1. Three patients with cancers were found homozygous for the R610del mutation while the R610del mutation is the most common mutation in patients with ASMD associated with an attenuated ASMD (NP-B) phenotype.
2. Do the Authors have other (than more severe disease course) explanations/possibilities about the association of the existence of cancer with larger spleen?
Author Response
We thank the reviewer for this review.
We understand the questioning related to the occurrence of cancer in the R610del mutation ASMD patients. Indeed, R610del is associated with a chronic visceral phenotype of ASMD. However, despite R610del homozygosity some patients suffer from a very severe form or the disease with respiratory failure and severe cytopenia. R610del mutation is not benign. It is difficult to study the incidence of cancer in neurologic ASMD (NP-A) due to the shorter life expectancy.
However, we have to admit that R610del homozygosity is generally associated with consanguinity, which could play a role in cancer development. In our small sample of patients, consanguinity was not specifically associated with a higher risk for cancer. A larger cohort could bring different results. We had this limit in our manuscript.
In our experience of physician in Internal Medicine, large spleen is not associated with cancer development. The spleen can be enlarged by cancer metastasis or an hemopathy by itself but splenomegaly is not known to be a risk factor for cancer development. Therefore we firmly believe that the large spleen in our patients is due to the ASMD severity and that severe ASMD is associated with cancer.